# Retrieval of a 3D CAD Model of a Transformer Substation Based on Point Cloud Data

Lijuan Long [1], Yonghua Xia [2,*], Minglong Yang [2], Bin Wang [2] and Yirong Pan [1]

[1] Faculty of Land Resources Engineering, Kunming University of Science and Technology, Kunming 650093, China
[2] Department of Earth Science and Technology, City College, Kunming University of Science and Technology, Kunming 650233, China
* Correspondence: 20040063@kust.edu.cn

**Abstract:** When constructing a three-dimensional model of a transformer substation, it is critical to quickly find the 3D CAD model corresponding to the current point cloud data from a large number of transformer substation model libraries (due to the complexity and variety of models in the model base). In response to this problem, this paper proposes a method to quickly retrieve a 3D CAD model. Firstly, a 3D CAD model that shares the same size as the current point cloud model bounding box is extracted from the model library by the double-layer bounding box screening method. Then, the selected 3D CAD model is finely compared with the point cloud model by the multi-view method. The 3D CAD model that has the highest degree of corresponding to the point cloud data is acquired. The proposed algorithm, compared to other similar methods, has the advantages of retrieval accuracy and high efficiency.

**Keywords:** point cloud data; CAD model; 3D model retrieval; 3D reconstruction of transformer substation

## 1. Introduction

In recent years, the electricity demand has increased in China, and more transformer substations have been built, renovated, and expanded. The traditional design and management of models cannot adapt to the development of substations. The intelligent management of substations has become a development trend. Therefore, building a substation 3D model is the foundation for realizing the intelligent management of transformer substations. Finding the 3D CAD model that corresponds to the point cloud data from the model base is key to the reconstruction of a substation's 3D model.

In engineering, during the process of building the 3D model of a substation, the most common steps include: building a 3D CAD model library of the transformer substation structures; finding a 3D CAD model that matches with the point cloud data from the model's library, according to the substation building point cloud data; as well as understanding the registration of the model and point cloud. However, 3D model libraries have large numbers of models with complex and diverse types; therefore, it takes a long time to find the matching model to the point cloud data of the substation construction in the library [1,2]. That is why 3D model automatic retrieval technology came about. OSADA R et al. [3] proposed a description method based on shape distribution features, by using geometric description operators D1, D2, D3, and A3 to calculate the point feature information of the 3D model surface; the best is the D2 shape operator. The D2 shape operator constructs a shape histogram by counting the Euclidean distance between any two points on the surface of the 3D model and retrieving the model by calculating the similarity of the shape histogram. This method is simple (regarding calculations) and has geometric invariances (regarding translating, scaling, and rotating the model), but its retrieval accuracy is low. Li Y.F. [4] improved the algorithm proposed by Osada to some

extent, using the distance (D2) of any two vertices in the model and the model based on the cosine value as two statistical features of the 3D model, using the correlation weighted feedback algorithm to determine the weight to combine the two geometric features and express the three-dimensional model's features. By this method, although the retrieval accuracy of the D2 algorithm is improved, the retrieval efficiency and stability are reduced. Bey A. et al. [5] used statistical methods to calculate the minimum fitting distance between the three-dimensional model and the point cloud data and determined the type of basic model according to the minimum matching distance between the basic model and the industrial point cloud data. Finally, the existing three-dimensional model has been truncated and assembled to match the point cloud data in an industrial environment. This method is mainly applicable to a three-dimensional model fitting, represented by the cylindrical model, but there are some problems involving data loss and unsmooth transitions in the reconstruction model. Ankerst et al. [6] proposed a model retrieval method based on the shape distribution histogram, which represents the 3D model by statistically measuring the radial distance of each vertex of the 3D model, and using sector segmentation, spherical shell segmentation, and a combination of the two for retrieval purposes. The advantages of this method include its simple calculation and fast speed, but the disadvantages are its poor robustness to the mesh subdivision and mesh simplification of the 3D model. To improve the retrieval accuracies of 3D models, Zhang K X et al. [7] proposed a semantic fusion method for the local structure retrieval of 3D CAD models. This method can better realize the local detailed retrieval of 3D CAD models through semantic fusion. However, this method is only applicable to the retrieval of the local similarity of models, and the retrieval efficiency is low.

To solve the problem of low efficiency and poor accuracy in finding the corresponding 3D CAD model from the model library, one must improve the efficiency of constructing the 3D model of the substation and reduce the labor and financial costs of the 3D model construction. This paper presents a method to quickly retrieve the corresponding 3D CAD model from the model library based on the 3D point cloud data. Firstly, we extracted the 3D CAD model in the model library, corresponding to the same size as the point cloud model by the double-layer bounding box (bounding sphere and OBB bounding box) screening method. Then, we used the multi-view method to filter the 3D CAD model to compare it with the point cloud model. Finally, we obtained the CAD model with the highest degree of matching with the point cloud data. Compared with the traditional methods, the method presented in this paper has the advantages of good retrieval accuracy and high efficiency. The research process is shown in Figure 1.

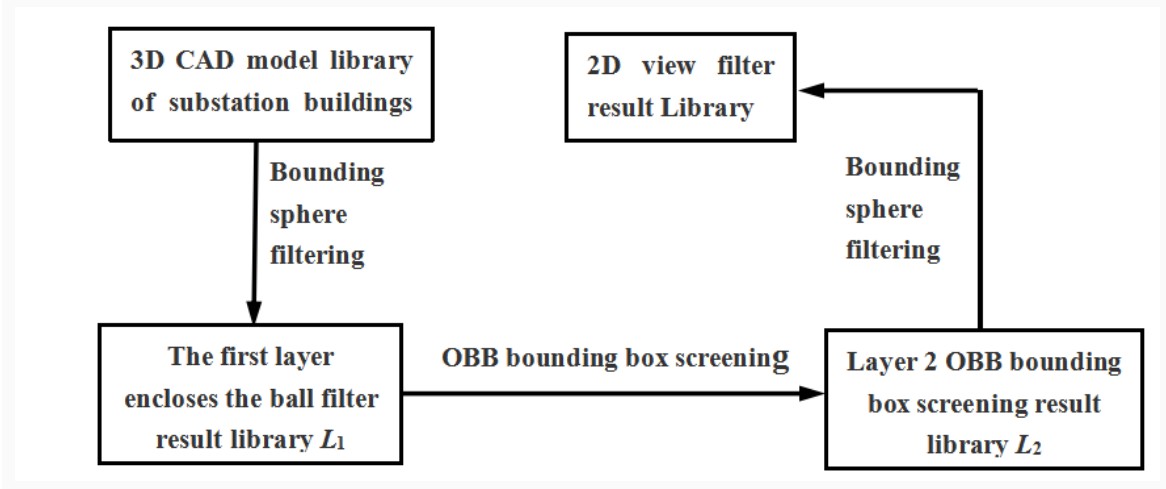

**Figure 1.** The 3D CAD model retrieval of transformer substation buildings based on point cloud data.

## 2. 3D CAD Model Retrieval

### 2.1. Screening of the Double-Layer Bounding Box Model Based on the Bounding Sphere and OBB Bounding Box

The bounding box is an optimal closed-space algorithm used for solving a set of discrete points. The commonly-used bounding box algorithms include a bounding ball, an axially-aligned bounding box (AABB bounding box), a directional bounding box (OBB bounding box), and so on.

The bounding ball is the smallest sphere that contains the object. When the object rotates, the bounding ball does not need to be updated and has rotational invariance. The AABB bounding box is the smallest hexahedron that contains the object. When the object rotates, it cannot rotate correspondingly and does not have rotational invariance. The OBB bounding box is insensitive to the object's direction and has an arbitrary direction, which enables it to surround the object as much as possible according to the shape characteristics of the surrounding object, and has rotational invariance. Because the attitudes of CAD models and the input point cloud data in transformer substation-building 3D model libraries are different, the bounding box of the model and point cloud data are required to have rotational invariance in the process of 3D CAD model screening. Bounding balls and OBB bounding boxes are the best choices as shown in the Figure 2.

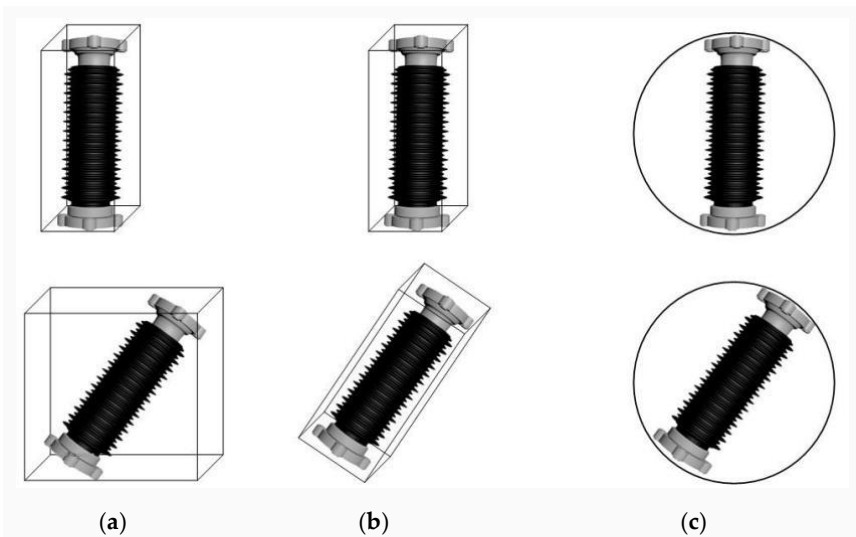

**Figure 2.** The bounding boxes of the objects. (**a**) AABB bounding box; (**b**) OBB bounding box; (**c**) bounding sphere.

The 3D CAD model retrieval technology studied in this paper involves searching for the input-specific 3D point cloud data in the model library to obtain a similar model. The advantages of using the OBB bounding box as the feature descriptor for 3D model retrieval is that it can retrieve a model of the same size as the existing point cloud from the model library, and provide data support for the later refined retrieval. However, the creation of the OBB bounding box is complex, and it takes a long time to retrieve. Compared with the OBB bounding box, the creation process of the bounding sphere is fast and the structure is simple. Using the bounding sphere as the feature descriptor can retrieve the CAD model from the model library that is consistent with the radius of the bounding sphere of the current point cloud. To obtain a 3D CAD model that shares the same size as the current point cloud data, further retrieval is required. To satisfy the demands of controlling the retrieval time while ensuring its accuracy, the method of combining the bounding sphere and the OBB bounding box is the most sensible choice. The first layer of rough screening is performed through the bounding sphere to reduce the number of CAD models retrieved in the library. Then, the OBB bounding box is used for further filtering to obtain a CAD model that is consistent with the size of the current point cloud.

(1)  The first-layer model screening (model screening based on the bounding sphere).

Before the model screening, firstly, the three-dimensional CAD model in the building model's substation library and the current point cloud data should be built separately to form the bounding ball. Then, the 3D CAD model is selected with the radius, consistent with that of the sphere surrounded by the point cloud data. This provides data support for the second-layer model screening. When constructing the bounding sphere of a 3D CAD model and point cloud data, we simply need to determine the center and radius of the sphere. We calculate the maximum and minimum values of the *X*, *Y*, and *Z* coordinates of the vertices of all elements in the set of basic geometric elements that make up the model (the vertices of the triangular patch of the 3D CAD model or the points of the point cloud data model), and make the mean values of the maximum and minimum values as the center of the ball. Then, we calculate the distance between the maximum and the minimum of *X*, *Y*, and *Z* as the diameters of the sphere [8].

Ball center:

$$O = \left( o_x, o_y, o_z \right) \tag{1}$$

$$o_x = \frac{1}{2}(x_{\max} + x_{\min}), \ o_y = \frac{1}{2}(y_{\max} + y_{\min}), \ o_z = \frac{1}{2}(z_{\max} + z_{\min}) \tag{2}$$

Radius:

$$R = \frac{1}{2}\sqrt{(x_{\max} - x_{\min})^2 + (y_{\max} - y_{\min})^2 + (z_{\max} - z_{\min})^2} \tag{3}$$

The equation of the bounding sphere:

$$(x - o_x)^2 + \left( y - o_y \right)^2 + (z - o_z)^2 = R^2 \tag{4}$$

$x_{\max}$, $y_{\max}$, $z_{\max}$ $x_{\min}$, $y_{\min}$, and $z_{\min}$ represent the maximum and minimum values of the model's vertex projection on the *x*, *y*, and *z* coordinates. *O* is the center of the bounding sphere. $o_x$, $o_y$, and $o_z$ are the components of the center coordinates of the bounding sphere. *R* is the radius of the bounding sphere. The steps of using the bounding sphere for the model screening are as follows:

When $\Delta\sigma \leq 0.1$ m, the bounding sphere of the current CAD model in the 3D model library is the same size as that of the point cloud data, and has similarities; thus, it is added to the first-layer screening result library (otherwise, the current model is not similar to the point cloud data). The 3D CAD models similar to 3D point cloud data are sequentially found from the 3D model library, which can provide data support for the second-layer OBB bounding box screening. This can greatly reduce the amount of data in the OBB bounding box screening, and improve the screening speed. Model screening process based on bounding sphere is shown in Figure 3.

(2)  The second layer of the model screening (model screening based on the OBB bounding box).

After the first-layer model screening, the 3D CAD models that are equal to the radius of the bounding sphere of the point cloud model of the substation building are selected. However, there are a large number of 3D CAD models in the model library that share the same radius as the surrounding sphere of the point cloud model of the substation building. This is not conducive to the comparison of the later model details. Therefore, based on the first-layer model screening, the second model screening is carried out to screen out 3D CAD models with similar sizes to the point cloud model, to reduce the amount of data for refined comparisons of later models, and to improve retrieval efficiency.

When constructing the OBB bounding box for the 3D CAD model in the first-layer screening result, the 3D CAD model is regarded as a complex structure composed of many triangular patches, so the vertices of the triangular patches provide data support for the creation of the OBB bounding box for the 3D CAD model. There are many algorithms used to construct the OBB bounding box. In this paper, the principal component analysis

(PCA) was used to obtain the three main directions of the triangle vertex to create the OBB bounding box [9]. The specific method is as follows: to obtain the vertex centroid of the triangle patch, calculate the covariance, construct the covariance matrix, and generate the eigenvalues and eigenvectors of the covariance matrix. The eigenvector is the main direction of the OBB bounding box.

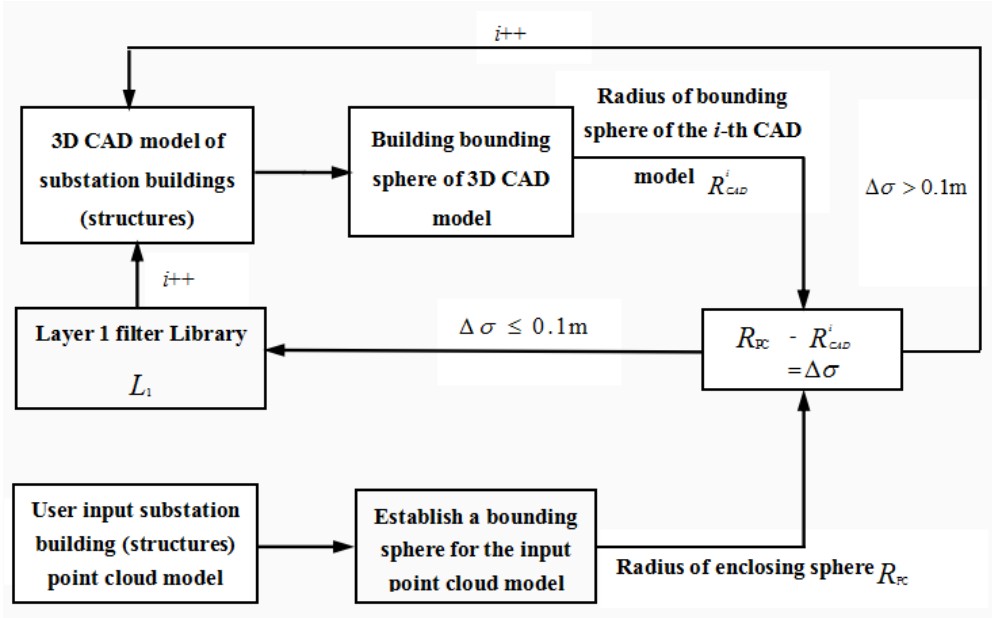

**Figure 3.** Model screening based on bounding sphere.

First, the mean value of the vertices of the triangular patch is calculated, where $n$ is the number of vertices.

$$\overline{x} = \frac{1}{n}\sum_{i=1}^{n} x_i \quad \overline{y} = \frac{1}{n}\sum_{i=1}^{n} y_i \quad \overline{z} = \frac{1}{n}\sum_{i=1}^{n} z_i \tag{5}$$

After obtaining the mean value of the vertices of the triangular patch, the covariance matrix $C$ of the vertices can be obtained.

$$\text{cov}(x,x) = \frac{1}{n-1}\sum_{i=1}^{n}(x_i - \overline{x})^2 \quad \text{cov}(x,y) = \frac{1}{n-1}\sum_{i=1}^{n}(x_i - \overline{x})(y_i - \overline{y}) \tag{6}$$

$$C = \begin{bmatrix} \text{cov}(x,x) & \text{cov}(x,y) & \text{cov}(x,z) \\ \text{cov}(y,x) & \text{cov}(y,y) & \text{cov}(y,z) \\ \text{cov}(z,x) & \text{cov}(z,y) & \text{cov}(z,z) \end{bmatrix} \tag{7}$$

The Jacobi iterative algorithm is used to solve the eigenvector of covariance matrix $C$ [10,11], and then the Schmidt orthogonalization algorithm is used to orthogonalize the eigenvector with the orthogonalized eigenvectors of $b_1$, $b_2$, and $b_3$ [12]. $b_1$, $b_2$, and $b_3$ are taken as the coordinate axes of the OBB bounding box. The $X$, $Y$, and $Z$ coordinates of the current 3D CAD model's triangular patch vertex are projected onto the calculated coordinate axes $b_1$, $b_2$, and $b_3$, respectively, to obtain the triangular patch vertex data in the new coordinate system.

$$(x',y',z') = \begin{pmatrix} b_1 \\ b_2 \\ b_3 \end{pmatrix}(x,y,z)^T \tag{8}$$

$(x',y',z')$ represents the vertex coordinates of the 3D CAD model in the $b_1$, $b_2$, $b_3$ coordinate system, and the origin of the $b_1$, $b_2$, $b_3$ coordinate system is o.

$$O = \left( \frac{1}{n} \sum_{i=1}^{n} x'_i, \frac{1}{n} \sum_{i=1}^{n} y'_i, \frac{1}{n} \sum_{i=1}^{n} z'_i \right) \qquad (9)$$

We obtained the maximum values of $x'$, $y'$, $z'$, $x'_{max}$, $y'_{max}$, $z'_{max}$, $x'_{min}$, $y'_{min}$, $z'_{min}$ in the $b_1$, $b_2$, $b_3$ coordinate system, respectively, and took $\max(|x'_{max}|, |x'_{min}|)$, $\max(|y'_{max}|, |y'_{min}|)$, and $\max(|z'_{max}|, |z'_{min}|)$ as the half length of the OBB bounding box to obtain the OBB bounding box of the 3D CAD model, which represents the absolute value.

The construction method of the OBB bounding box of the 3D point cloud model is the same as that of the 3D CAD model, which will not be described in detail here.

After building the OBB bounding box for the point cloud data model and the 3D CAD model, the 3D CAD model of the same size as the current OBB bounding box for the 3D point cloud data is selected from the first-layer screening result library ($L_1$). The process is shown in Figure 4, where $S_{PC}$ represents the area of the OBB bounding box of the 3D point cloud of the substation building, $S_{CAD}$ represents the area of the OBB bounding box of the CAD model in $L_1$, $\Delta\sigma$ = 0.1 m. $Hl^{PC}_{max}$ and $Hl^{PC}_{min}$ represent the maximum and minimum half-lengths of the OBB bounding box of the current 3D point cloud data. $Hl^{CAD}_{max}$ and $Hl^{CAD}_{min}$ represent the maximum and minimum half-lengths of the OBB bounding box of the CAD model in $L_1$.

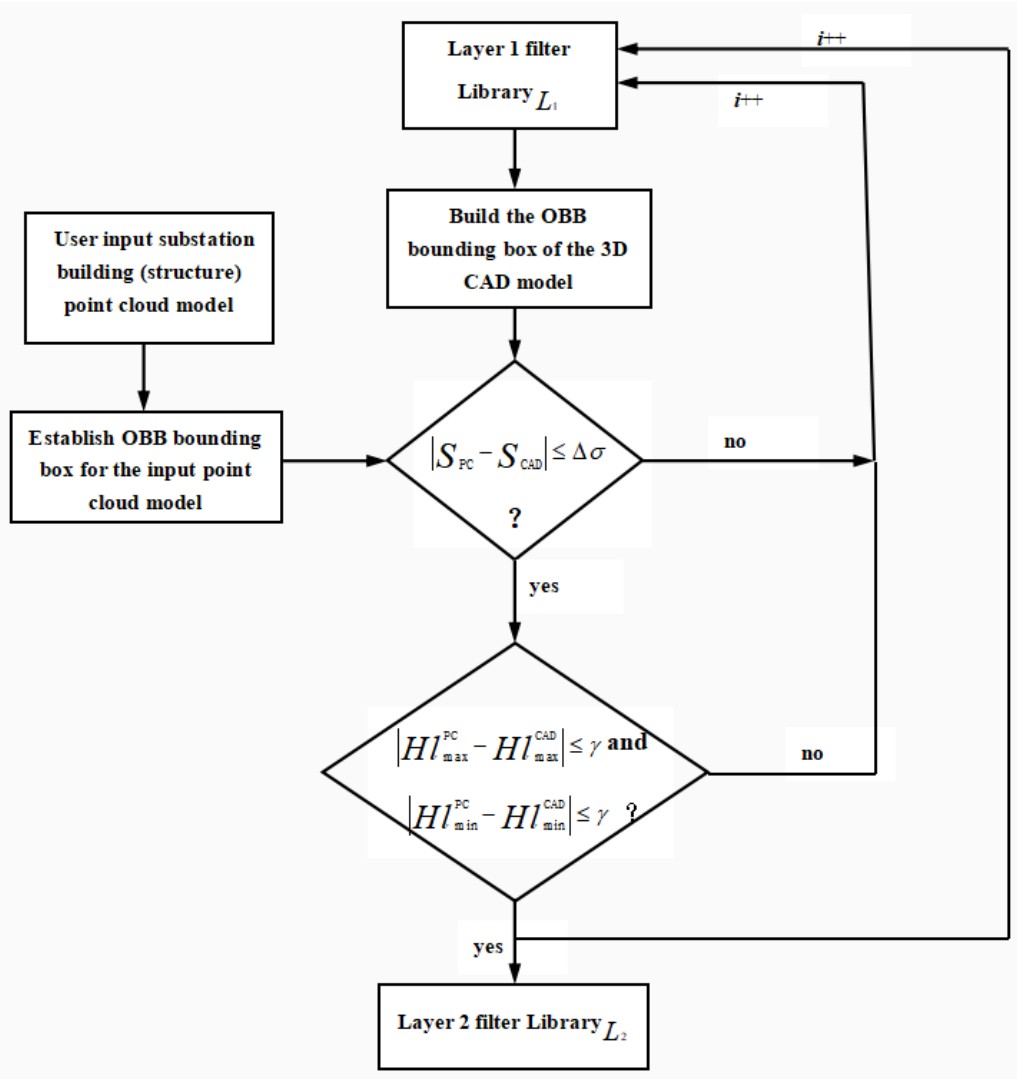

**Figure 4.** Model screening based on the bounding sphere.

## 2.2. Detailed Retrieval of the 3D CAD Model

The double-layer bounding box model screening process, based on the bounding sphere and OBB bounding box, belongs to the similarity screening stage of the global features of the 3D CAD models in the model library (using the point cloud data model). This method ignores the local features of the model and lacks the ability to distinguish the local details of the model. The bounding box screening method can only filter out 3D CAD models with similar appearances to the point cloud data model from the model library. Some of these models may only have similar external sizes to the point cloud data, while the local details are very different from the point cloud data. Therefore, after the preliminary screening of the bounding box, it is necessary to refine the local details of the model to screen out a 3D CAD model that is more similar to the current point cloud data. Thus, this paper adopted a view-based 3D model retrieval method on the basis of previous research results.

A view-based 3D model retrieval algorithm is a content-based 3D retrieval method that can significantly reduce the complexity of the algorithm [13]. The core concept is to transform the existing model into a set of two-dimensional images through a group of virtual cameras and then obtain the model similarity by calculating the distance between the two-dimensional images [14].

(1)     Generate a 2D image of the model.

The view-based 3D model retrieval algorithm is difficult for such methods to balance the efficiency of the 3D model retrieval process and the accuracy of the retrieval results. The accuracy of the view-based 3D model retrieval results is proportional to the detailed description of the 3D model features, i.e., related to the number of 2D images. The storage of 2D images requires a large space. The comparison of images involves computational costs, which reduce the retrieval efficiency. Therefore, when collecting the 2D images of 3D models, we should use algorithms that are relatively simple and can fully describe the details of 3D models [15]. Considering the retrieval time and efficiency, it is more appropriate to use the hexahedron-based view acquisition method proposed by Shih et al. [16] when acquiring 2D views of 3D models. Firstly, a regular hexahedron with the same centroid as the model is constructed, and the side length of the hexahedron is *L*. Then, the model is mapped to six faces corresponding to the hexahedron. Finally, the corresponding features are extracted to match the similarity between the models. Two-dimensional image of the model obtained by the means of Shih et al. is shown in Figure 5.

$$L = 2\max\left\{ \sqrt{((x_i - \overline{x})^2 + (y_i - \overline{y})^2)} \,\middle|\, i = 0, 1, 2 \ldots n \right\} \tag{10}$$

(2)     Extraction of image features

After obtaining 2D views of the point cloud model and 3D CAD model, extracting 3D model features from these views is an important step in the view-based 3D model retrieval algorithm. In this paper, we adopted the Canny operator to extract the edge features of different view models. The algorithm for using the Canny operator to extract the edge features is as follows:

① We used Gaussian filtering to smooth the image. Let $g(x, y)$ be the original image, $f(x, y)$ the smoothed image, and $G(x, y)$ be the Gaussian kernel function, then:

$$G(x, y) = \frac{1}{\sqrt{2\pi\sigma^2}} e^{\frac{-(x^2 + y^2)}{2\sigma^2}} \tag{11}$$

$$f(x, y) = g(x, y) \times G(x, y) \tag{12}$$

② The gradient value and gradient direction are calculated. The core concept of the Canny operator is to find the position where the gray intensity changes the most in the image. The so-called maximum change is the gradient direction. We used the finite

difference of the first-order partial derivative to calculate the magnitude and direction of the $f(x)$ gradient.

$$f_x = \frac{1}{2}[f(x+1,y) - f(x,y) + f(x+1,y+1) - f(x,y+1)] \tag{13}$$

$$f_y = \frac{1}{2}[f(x,y+1) - f(x,y) + f(x+1,y+1) - f(x+1,y)] \tag{14}$$

$$M = \sqrt{(f_x^2 + f_y^2)} \tag{15}$$

$$\theta = \arctan\frac{f_y^2}{f_x^2} \tag{16}$$

$M$ is the intensity of the image edge and $\theta$ is the direction of the image edge.

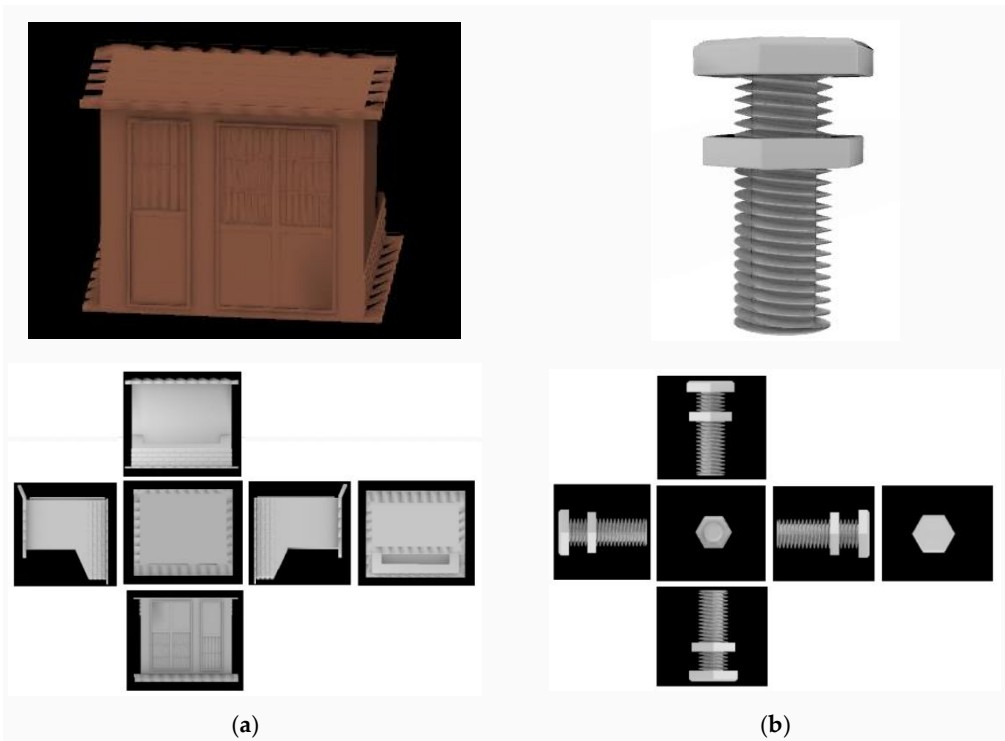

        (**a**)               (**b**)

**Figure 5.** Two-dimensional image of the model obtained by the means of Shih et al. [16]. (**a**) Garbage room, (**b**) screw.

 ③ Filter non-maxima. The gradient intensity value of the current pixel is compared with two pixels in the positive and negative directions. If the current pixel value is greater than these two pixel values, the current pixel is the edge point; otherwise, the current pixel is suppressed, i.e., the current pixel is assigned a value of 0.

 ④ After filtering the non-maximum value, the image edge can be more accurately represented by the remaining pixels. However, some edge pixels are inevitably caused by noise and color changes. In order to solve this problem, we can use weak gradient values to filter the edge pixels, while retaining the edge pixels with high gradient values. If the gradient value of the edge pixel is greater than the high threshold, it is marked as a strong edge pixel. On the contrary, if the gradient value of the edge pixel is less than the low threshold, the edge pixel will be suppressed. The specific implementation is shown in Algorithm 1:

**Algorithm 1**: Filtering the Edge Pixels

| |
|---|
| 1:    If $f_p$ > Height Threshold && $f_p$ = Height Threshold |
| 2:        $f_p$ is a strong edge |
| 3:    else if $f_p$ > Low Threshold && $f_p$ = Height Threshold |
| 4:        $f_p$ is a weak edge |
| 5:    else |
| 6:        $f_p$ should be Suppressed |

⑤ Suppress isolated low threshold points.

So far, the pixels extracted from the real edge of the image have been identified as edges. However, for some weak edge pixels, there are still some disputes, because it is uncertain whether these pixels are extracted from the real edge, or caused by noise or color changes. By suppressing the interference of weak edge pixels caused by noise or color changes, the accuracy of the obtained results can be guaranteed. Generally, there is a certain connection between the weak edge pixels and the strong edge pixels caused by real edges, but there is no connection between the two caused by the noise points. Therefore, we can check the eight adjacent pixels of the weak edge pixels. If at least one of them is a strong edge pixel, the weak edge pixels can be retained as the real edge. Otherwise, the weak edge pixels are suppressed. The specific implementation is shown in Algorithm 2:

**Algorithm 2**: Suppress Isolated Low Threshold Points

| |
|---|
| 1:    If $f_p$ == Low Threshold && $f_p$ connected to a strong edge pixel |
| 2:        $f_p$ is a strong edge |
| 3:    else |
| 4:        $f_p$ should be Suppressed |

The canny edge feature is shown in Figure 6. Canny edge descriptors are extracted from six images of the model to acquire the edge information of six angles of the model. Suppose $K_c$ represents each edge information and $K_C$ represents the features of the 3D model [17–19], namely:

$$K_C = [k_{c1}, k_{c2}, k_{c3}, \ldots, k_{cn}] \tag{17}$$

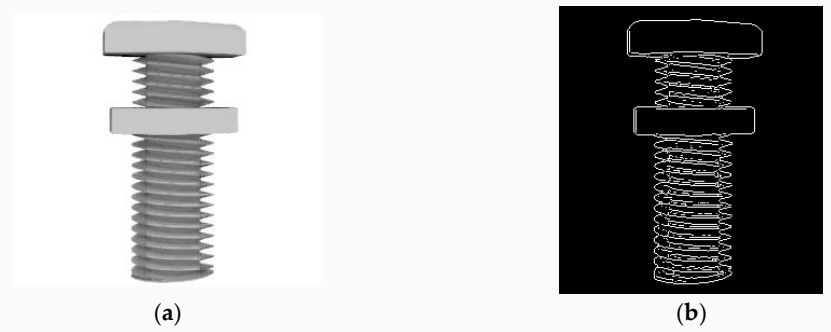

(a)

(b)

**Figure 6.** Canny edge feature. (**a**) Screw projection view, (**b**) screw edge extraction map.

### 2.3. Model Similarity Calculation

The purpose of the model similarity calculation is to calculate the similarity between model feature descriptors. The smaller the distance deviation between the feature descriptors of the different models, the more similar the models are, and vice versa. There are many distance algorithms used to quantify model similarity, such as Euclidean distance, Hausdorff distance, Manhattan distance, cosine similarity algorithm, and so on. In this paper, Euclidean distance, the most commonly used method for distance, was adopted to calculate the similarity between the model in the model library and the current point cloud data model.

Let $K_A = \{k_{a1}, k_{a2}, k_{a3}, \ldots, k_{an}\}$ denote the feature vector of 3D point cloud model *A* entered by users, and $K_B = \{k_{b1}, k_{b2}, k_{b3}, \ldots, k_{bn}\}$ denote the feature vector of 3D CAD model *B* in the model library. $D(A, B)$ represents the distance between models *A* and *B*. The distance calculation formula is as follows:

$$D(A, B) = \sqrt{\sum_{i=1}^{n} (k_{ai} - k_{bi})^2} \tag{18}$$

## 3. Experimental Result

The model library used in this paper is a self-built substation model library. There are 1600 3D CAD models of substation buildings (structures), as shown in Figure 7. It includes substation models of 20, 110, 220, 330, and 500 kV. During the reconstruction of the substation, it takes a lot of manpower to find a substation model similar to the current point cloud data model from 1600 models. Therefore, this paper proposes a method to quickly retrieve the 3D model from the model library based on the point cloud data model. In order to verify the feasibility of the above algorithm, we used MATLAB 2019 to verify it.

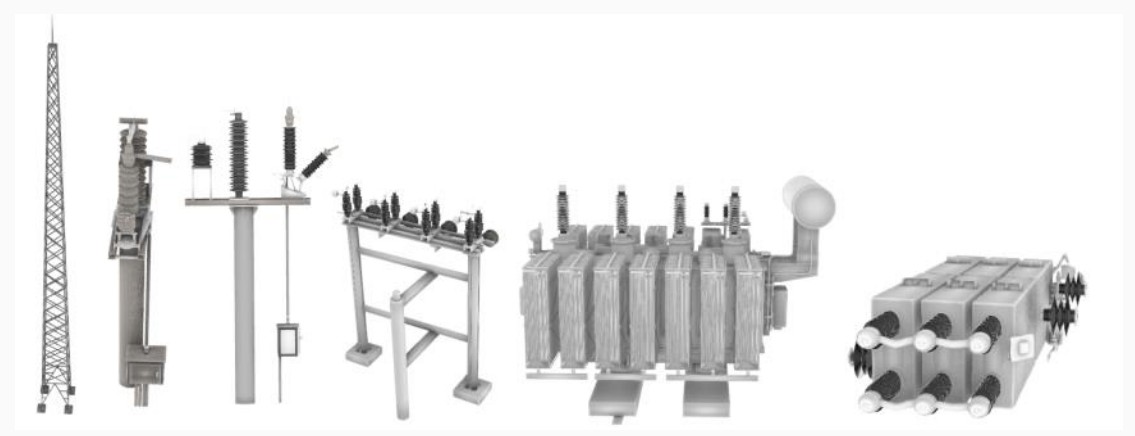

**Figure 7.** Part of a 3D CAD model in the substation building model base.

### 3.1. Point Cloud Data Collection and Preprocessing

When retrieving the 3D model, the 3D point cloud data of the substation construction (building) is essential. In this paper, the I Site8200ER 3D laser scanner produced by the Australian company MAPTEK was used as the scanning instrument to collect point cloud data.

Due to the complexity and particularity of the substation scanning scene, the occlusion between electrical equipment, the environment of the surrounding electromagnetic field, and the limitation of the field of view of the 3D laser scanner, it is necessary to scan in multiple directions and angles to obtain multiple viewpoints. Multi-period point cloud data and noise are easily generated during scanning operations, resulting in a large amount of data redundancy, which is not conducive to later model retrieval. Therefore, before 3D model retrieval, the collected point cloud data should be preprocessed [20], i.e., point cloud data registration [21], point cloud data denoising [22], and streamlining [23,24]. Figure 8 shows the collection and processing of data.

### 3.2. 3D Model Retrieval Based on the Bounding Sphere and OBB Bounding Box

According to the point cloud data model, the bounding sphere is firstly used to screen the 3D CAD models of the substation buildings (structures) for initial screening, to reduce the number of models in the model library, and provide data support for later OBB bounding box screening. The screening results are shown in Table 1.

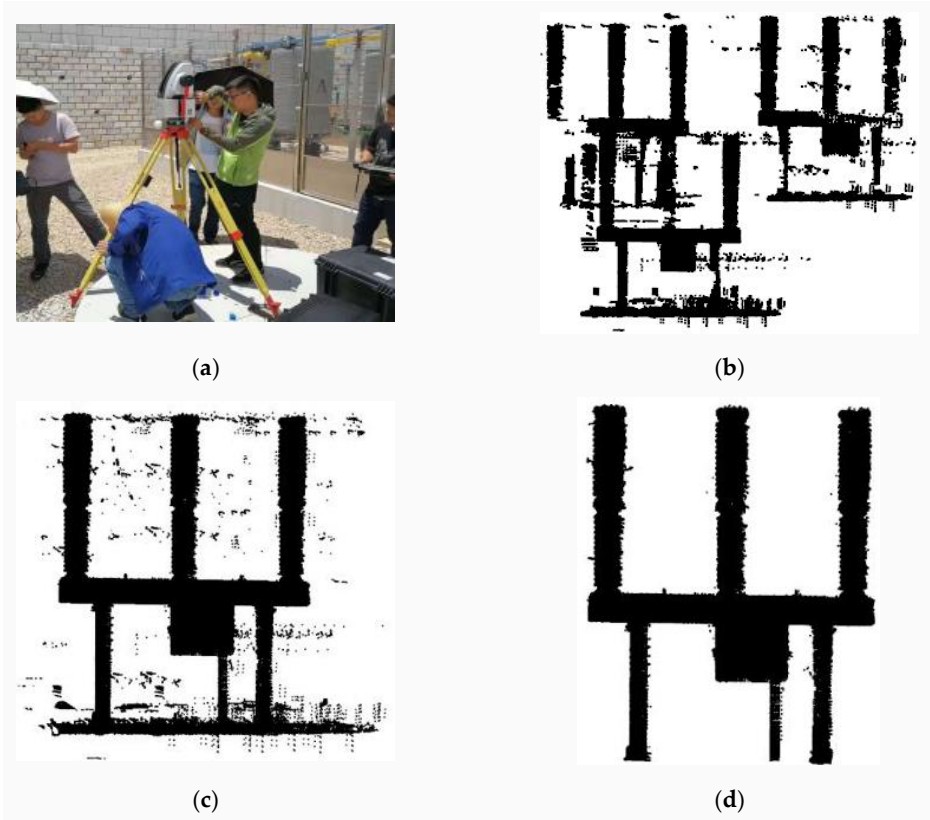

**Figure 8.** Collection and processing of point cloud data. (**a**) Collection of point cloud data. (**b**) Original point cloud data. (**c**) Registration of point cloud data. (**d**) Point cloud data after preprocessing.

**Table 1.** Results of the model screening by the bounding sphere.

| Point Cloud Model to Be Retrieved | 1 | 2 | 3 | ... |
|---|---|---|---|---|
| 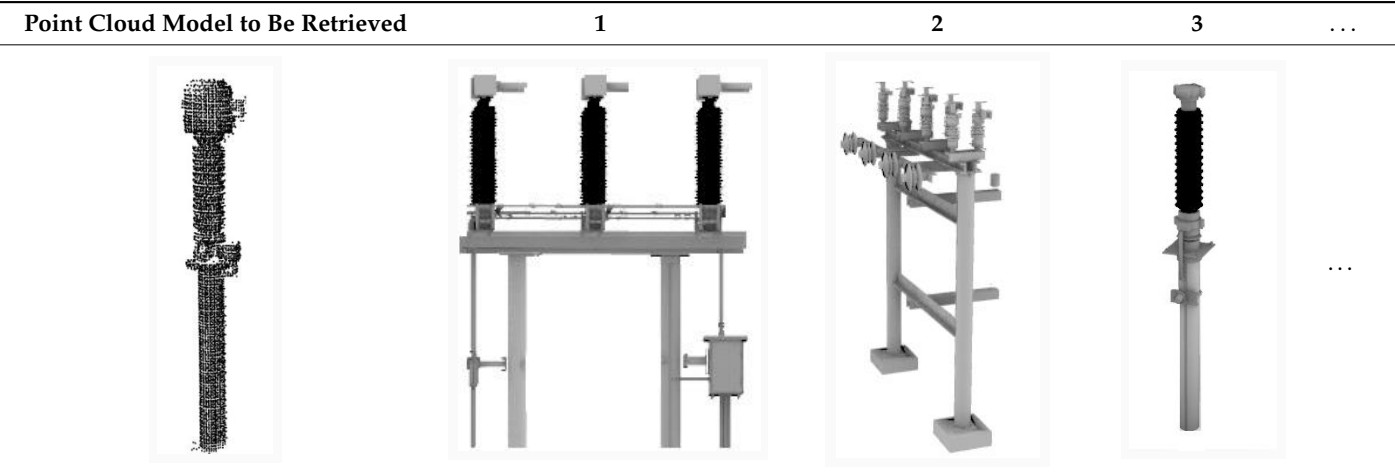 | | | | ... |

After completing the above process, the OBB bounding box is used to further screen the models in the model library, to reduce the number of candidate models, and improve the retrieval efficiency. The screening results are shown in Table 2.

### 3.3. 3D Model Retrieval Based on an Image

Our experiment uses the sunny operator to extract the features of the model, the results are shown in Figure 9, with the Gaussian radius of a Canny of 2, a low threshold of 40, and a high threshold of 120. The algorithm proposed in this paper, the D2 algorithm,

and the model screening results of the two-dimensional image proposed by Shih et al., as shown in Table 3.

**Table 2.** Results of the model screening by the OBB bounding box.

| Point Cloud Model to Be Retrieved | 1 | 2 | 3 | 4 | . . . |
|---|---|---|---|---|---|

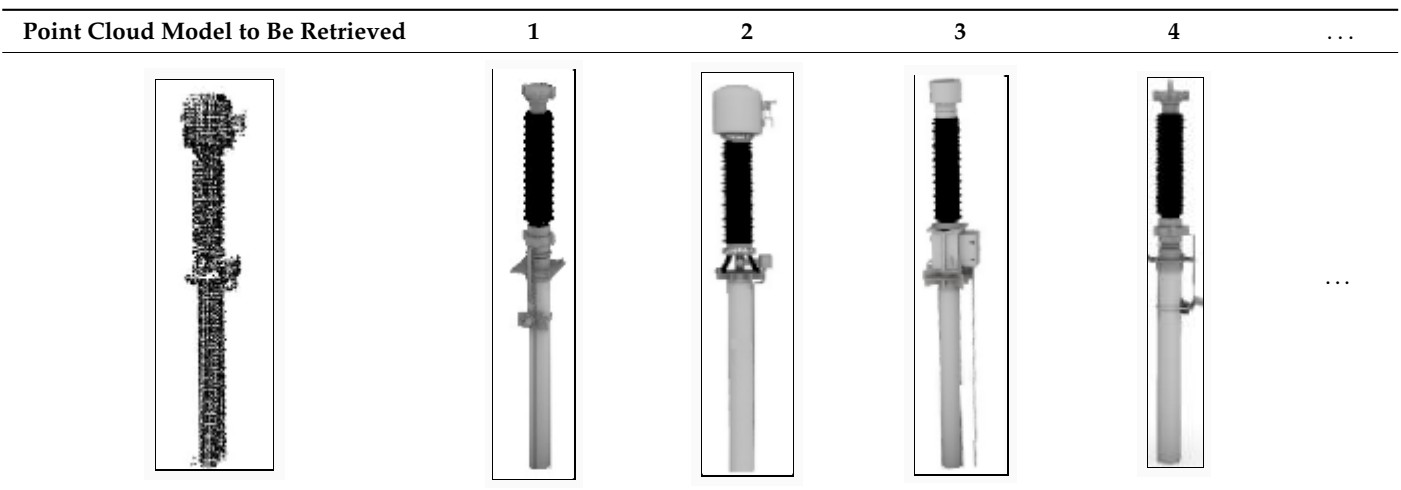

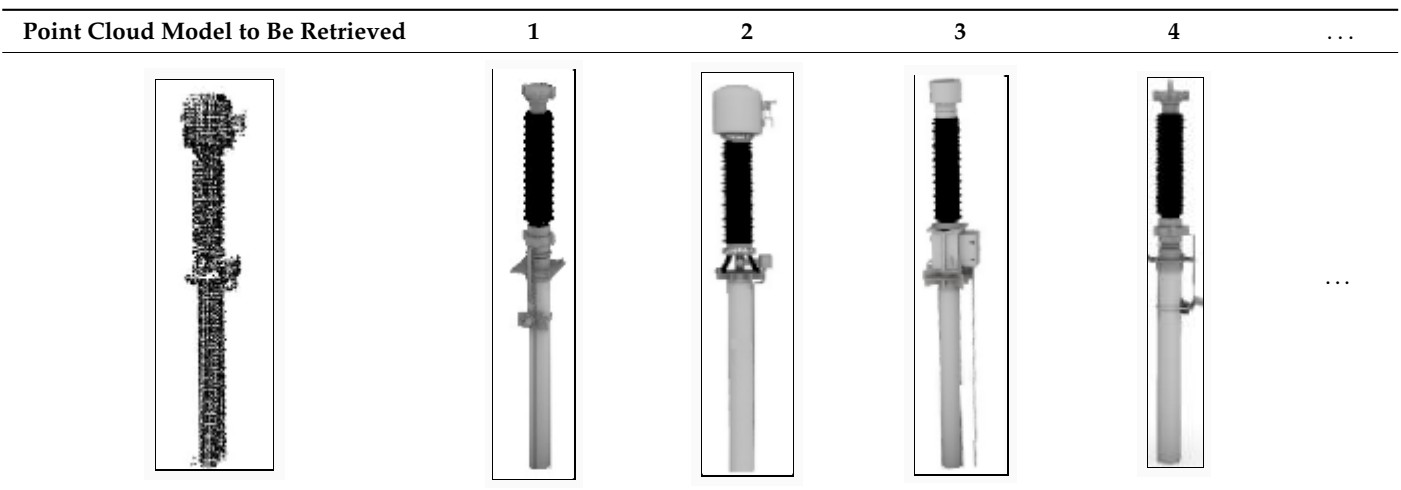

**Figure 9.** *Cont.*

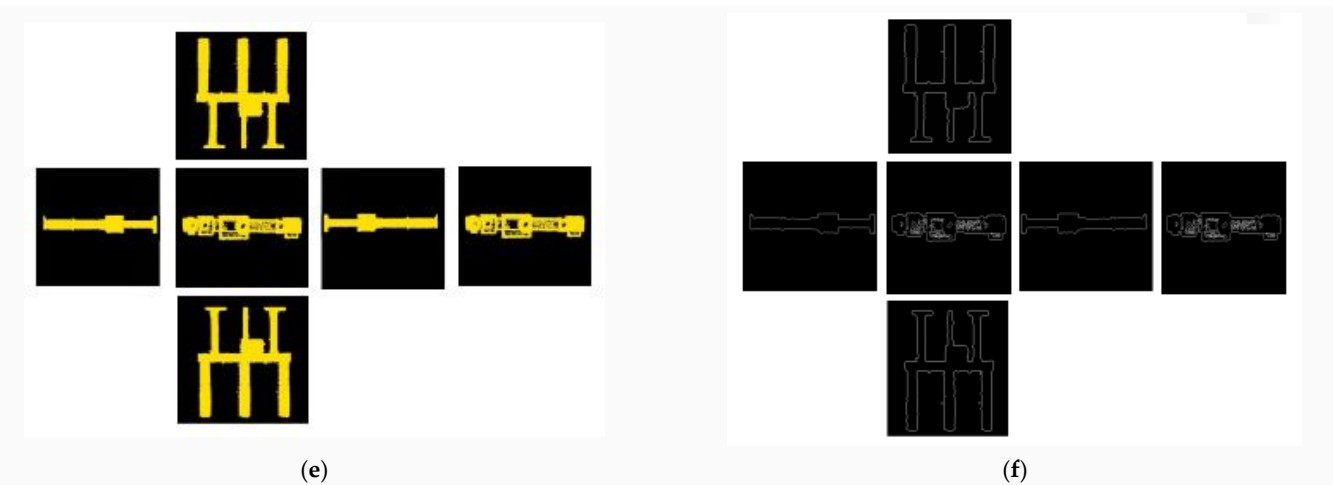

**Figure 9.** Edge feature of Canny. (**a**) The 3D CAD model of the voltage transformer; (**b**) 3D point cloud model of the voltage transformer. (**c**) Projected view of the voltage transformer; (**d**) edge extraction diagram of the voltage transformer; (**e**) point cloud projection view of the voltage transformer; (**f**) edge extraction of the Pt point cloud.

### 3.4. Analysis of Retrieval Accuracy

To evaluate the algorithm, the recall–precision ratio was used for verification. The recall rate is expressed by the ratio of the detected relevant CAD models to the total number of relevant CAD models in the retrieval system, i.e., the recall rate = (the retrieved relevant CAD models/the total number of relevant CAD models in the system) × 100%. The precision rate is expressed by the ratio of the detected related CAD models to the total number of detected CAD models, i.e., the precision rate = (total number of detected CAD models/relevant CAD models) × 100%.

It is evident from Figure 10 and Table 4 that the recall–precision ratio of the D2 algorithm is average, but the retrieval takes a long time on average. Although the algorithm proposed by Shih et al. takes less time to retrieve the model on average, its recall–precision ratio is poor. The recall–precision ratio of the 3D model retrieval by only using a double-layer bounding box screening method is the worst because the 3D model retrieval method based on the double-layer bounding box can only retrieve the CAD model with the greatest similarity to the current 3D model shape from the model library; the details of the model cannot be taken into account, but the double-layer bounding box filtering method can reduce the number of models in the model library. Only the 2D view retrieval algorithm proposed in this paper is used for the 3D model retrieval, which performs better in the recall–precision ratio, but the retrieval takes the longest time on average. Because the storage for 2D images is large and the comparison time for 2D images is long, this method is not recommended for model retrieval alone if the model library is large. By combining the two-layer bounding box filtering method with the two-dimensional view retrieval method (the algorithm proposed in this paper), the recall–precision ratio of the proposed algorithm proposed in this paper is better than other algorithms, and the average retrieval time is shorter. Therefore, only by combining the double-layer bounding box filtering method with the two-dimensional image retrieval algorithm can the recall and precision rates of the retrieval algorithm be optimized in shorter times.

**Table 3.** The algorithm proposed in this paper, the D2 algorithm, and the model screening results of the two-dimensional image proposed by Shih et al. [16].

| Point Cloud Model to Be Retrieved | Algorithm | 1 | 2 | 3 | 4 | . . . |
|---|---|---|---|---|---|---|
|  | The algorithm proposed in this paper |  |  |  |  | . . . |
| | View retrieval method proposed by Shih et al. |  |  |  |  | . . . |
| | D2 algorithm |  |  |  |  | . . . |

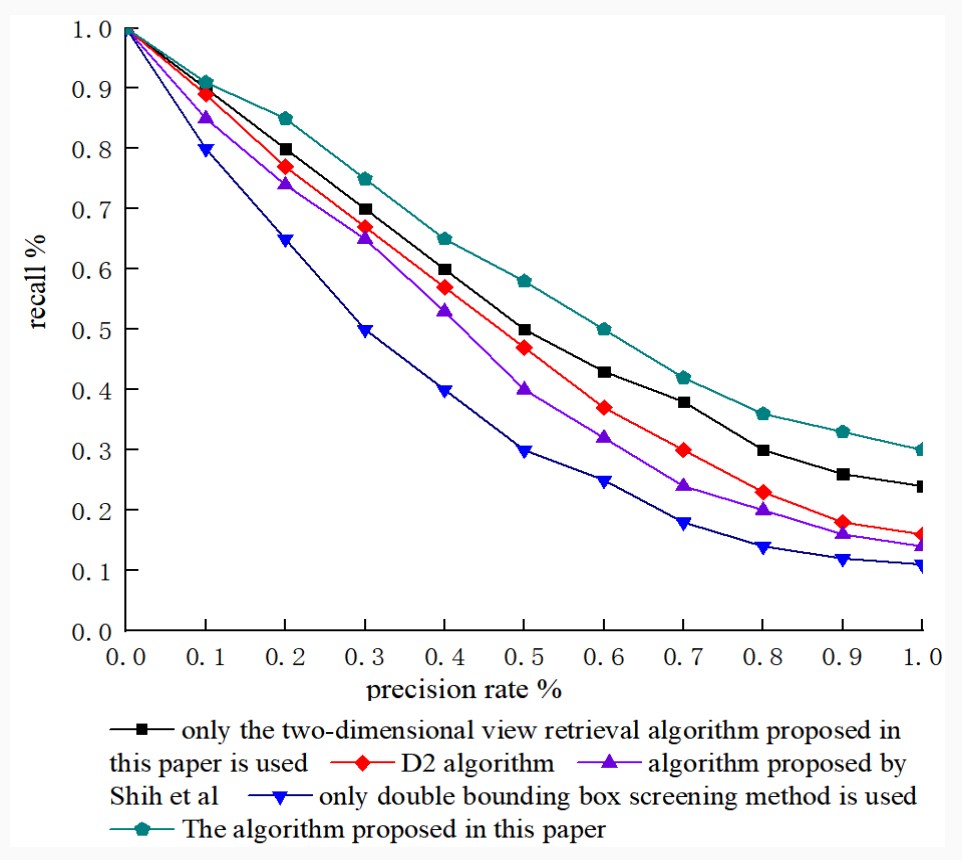

**Figure 10.** The proposed algorithm is compared with other algorithms.

**Table 4.** The time of the CAD model corresponding to the current point cloud data retrieved from the model base.

| Use Algorithm | D2 Algorithm | Algorithm Proposed by Shih et al. | Only the 2D View Retrieval Algorithm Proposed in This Paper is Used | The Algorithm Proposed in This Paper |
|---|---|---|---|---|
| average time spent (ms) | 3.165 | 3.130 | 3.249 | 3.158 |

## 4. Conclusions

In the process of substation modeling based on point cloud data, it is necessary to find the 3D CAD model corresponding to the current point cloud data from the model library; the efficiency of manual retrieval is low. Therefore, in this paper, we propose a fast automatic retrieval method for 3D models. First, a CAD model of a similar size to the OBB bounding box of the current point cloud data from the library must be found, then a multi-view-based method is used to compare the details between the models, and finally, the 3D CAD model corresponding to the point cloud data is obtained. We also compared the recall rate, precision rate, and retrieval efficiency with the existing fast retrieval methods, and concluded that the algorithm proposed in this paper has better accuracy and efficiency in retrieving 3D models of point cloud data.

**Author Contributions:** Conceptualization, L.L. and Y.X.; methodology, L.L.; validation, L.L., Y.X. and M.Y.; formal analysis, B.W.; resources, Y.X.; data curation, L.L. and Y.P.; writing—original draft preparation, L.L.; writing—review and editing, Y.P.; visualization, L.L. All authors have read and agreed to the published version of the manuscript.

**Funding:** This research received no external funding.

**Institutional Review Board Statement:** Not applicable.

**Informed Consent Statement:** Not applicable.

**Data Availability Statement:** Not applicable.

**Conflicts of Interest:** The authors declare no conflict of interest.

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
