# Peer review of "Retrieval of a 3D CAD Model of a Transformer Substation Based on Point Cloud Data"

_2673-4052, doi:10.3390/automation3040028_

Round 1
Reviewer 1 Report
The authors propose an interesting model. Their work is interesting but has some weaknesses.
In the introductory chapter, the authors try to justify the necessity of their research. At the beginning of the introduction, the authors mention other research in the field. At the end of the introductory chapter, the authors provide a very brief rationale of the research. Unfortunately, the authors avoid clearly presenting the objectives of their research.
Some graphics are used that are not understood where they come from, for example Figure 1. Is this the authors' proposal or not?
In chapter 2 the authors try to present their proposed model, but in this chapter the authors insist too much on other existing models. It would be more useful for the proposed model to be the central element of the description. Perhaps merging with chapter 3 would be appropriate in a better theoretical presentation of the proposed model.
Chapter 4 is dedicated to data and its processing. The authors avoid presenting details about the data collected. For this reason, both the data collected, and the results of their processing become unconvincing to support the conclusions, even if these conclusions are extremely few. Moreover, the conclusions have an insignificant personal imprint of the authors. In addition, the authors try to mention a weakness of the proposed method, but without sufficiently arguing this weakness.
Many bibliographic references are not in the required format and therefore difficult to consult. For example, the reference in position 5, which should normally be cited as: Liu, AA., Zhou, HY., Li, MJ. et al. 3D model retrieval based on multi-view attentional convolutional neural network. Multimed Tools Appl 79, 4699–4711 (2020). https://doi.org/10.1007/s11042-019-7521-8. The examples could go on, but I don't want to go through all the references here. The authors need to improve this part of the paper as well.
Reviewer 2 Report
Interesting article about: “Finding the corresponding 3D CAD model from the 3D Model Library based on the geometric features of the point cloud has become the research focus of substation reverse modeling.” The introduction contains a very short literature review (only 5 references on related work, but not on point clouds data-handling), and then the authors continues with their method: “To solve these problems, this paper presents method to quickly retrieve the corresponding 3D CAD model from the model library based on the 3D point cloud data.”
This reviewer is not an expert this field of 3D CAD model retreivel, but for sure this topic has been addressed in (related) point cloud research. See, for example:
Corsia, Mathias & Chabardès, T. & Bouchiba, H. & Serna, Andrés. (2020). LARGE SCALE 3D POINT CLOUD MODELING FROM CAD DATABASE IN COMPLEX INDUSTRIAL ENVIRONMENTS. ISPRS - International Archives of the Photogrammetry, Remote Sensing and Spatial Information Sciences. XLIII-B2-2020. 391-398. 10.5194/isprs-archives-XLIII-B2-2020-391-2020.
That paper contains a lot of other related references.
Without a proper “related work” session, addressing existing methods and techniques in retrieving corresponding 3D CAD models from a model library based on 3D point cloud data, the added value of the method describes in this paper could not be judged.
That said, the description of the method is very clear.
Round 2
Reviewer 1 Report
I have no further comments to make.
Author Response
Thank you very much. In the second round of revision, the author mainly revised the conclusions of the paper. Some adjustments have been made to the grammar problems.
Conclusion:
In the process of substation modeling based on point cloud data, it is necessary to find the 3D CAD model corresponding to the current point cloud data from the model library, and the efficiency of manual retrieval is low. Therefore, the authors propose a fast automatic retrieval method for 3D models in this paper. First, they find a CAD model of similar size to the OBB bounding box of the current point cloud data from library, then use a multi-view-based method to compare the details between the models, and finally obtain the 3D CAD model corresponding to the point cloud data. The authors also compares the recall rate, precision rate and retrieval efficiency with the existing fast retrieval methods, and concludes that the algorithm proposed in this paper shows better accuracy and efficiency to retrieve 3D models of point clouds data.
Reviewer 2 Report
Paper has been improved; the most by responding to the comments of the other reviewer.
I made this remark: "Without a proper “related work” session, addressing existing methods and techniques in retrieving corresponding 3D CAD models from a model library based on 3D point cloud data, the added value of the method describes in this paper could not be judged."
Still have some doubts.
Author Response
Review:I made this remark: "Without a proper “related work” session, addressing existing methods and techniques in retrieving corresponding 3D CAD models from a model library based on 3D point cloud data, the added value of the method describes in this paper could not be judged."
Still have some doubts.
Author: In view of the experts' problems, the author made some modifications to the conclusions of the paper.
Conclusion:
In the process of substation modeling based on point cloud data, it is necessary to find the 3D CAD model corresponding to the current point cloud data from the model library, and the efficiency of manual retrieval is low. Therefore, the authors propose a fast automatic retrieval method for 3D models in this paper. First, they find a CAD model of similar size to the OBB bounding box of the current point cloud data from library, then use a multi-view-based method to compare the details between the models, and finally obtain the 3D CAD model corresponding to the point cloud data. The authors also compares the recall rate, precision rate and retrieval efficiency with the existing fast retrieval methods, and concludes that the algorithm proposed in this paper shows better accuracy and efficiency to retrieve 3D models of point clouds data.